# Learning to Classify Contaminated Blood Tests: A Neural Approach

**Jason V. Mancuso, Peter Short & Dr. Elena A. Manilich**
Department of Mathematics and Computer Science
John Carroll University
Cleveland, OH 44118, USA
{jvmancuso87@gmail.com}

**Dr. Edmunds Z. Reineks**
Robert J. Tomsich Pathology &
Laboratory Medicine Institute
Cleveland Clinic
Cleveland, OH 44195 USA

## Abstract

Automated laboratories analyze thousands of blood tests every day. Some samples can become contaminated by improper collection, producing incorrect results. Identification of such samples is critical and requires a labor intensive expert review. In our previous work, we demonstrated a useful machine learning approach to automated detection, but a severe class imbalance in the dataset led to unstable results. Using a modified data augmentation strategy and an attentional neural architecture, we show significant improvement over previously published methods in all binary classification metrics, with an absolute F1-score improvement of 13%. This work represents the state-of-the-art in automated contamination detection and demonstrates that a machine learning pipeline based on our neural architecture can perform on par with human experts.

## 1 Introduction

A recent study from Najat (2017) at the Sulaimani diagnostic labs in Iraqi Kurdistan showed 70% of blood testing errors occur before the samples ever reach a laboratory. In particular, errors due to contamination are common and can be difficult to detect, as the severity of the contamination can vary drastically depending on its source. This can put patients at unnecessary risk of harm (Sharratt et al., 2009). Laboratory physicians can be trained to identify these contaminations manually from test results, but the volume of tests per day make it infeasible to do so for labs with automated chemistry systems. Of the methods that have been proposed to detect contaminated blood tests automatically, a rules-based approach has been most prevalent, while a machine learning approach has shown the most promise.

### 1.1 Related Work

Labs often use a rule-based algorithm for determining whether a blood test has been significantly skewed by a contamination. Although this class of algorithms is easily implemented, they generally perform poorly with respect to recall in this setting, which we experimentally validate by using a rules-based algorithm based on published guidelines (Hernandez, 2011) as a benchmark in Section 3.

Learning algorithms have recently been applied to this problem in Baron et al. (2012) and Mancuso et al. (2017). Baron et al. used the CART decision tree algorithm for classification. By using CART, the authors extend the rules-based method to include rules learned from data algorithmically. Their feature set included the eight analytes of the basic metabolic panel and a 30-day mean glucose measurement. They achieve reasonable performance with a small dataset ('above 80%' sensitivity and specificity), but the requirement of 30 day mean glucose feature makes this infeasible for general patient populations. Without this feature, the CART approach was not successful for them. We evaluate a similar CART-based decision tree to establish how these results scale to our larger dataset in Section 3.

While this work seems to suggest that the mean glucose over time feature is critical for classifying contaminated basic metabolic panels, Mancuso et al. (2017) shows that their result is most likely due

---

**Algorithm 1** Data augmentation for binary classification

---

**Given:** training set $D = \{(\mathbf{x}_i, y_i) | \mathbf{x}_i \in X, y_i \in Y\}$, number of mixture model components $k$,
model $\mathcal{M}$
Let $X_{pos} = \{\mathbf{x}_i | y_i = 1\}$ and $X_{neg} = \{\mathbf{x}_i | y_i = 0\}$.
Fit mixture model $p_k(\mathbf{x}|\theta)$ on $Z \in \{X_{pos}, X_{neg}\}$ such that $|Z| = \min(|X_{pos}|, |X_{neg}|)$.
Draw $\mathrm{abs}(|X_{neg}| - |X_{pos}|)$ samples from $p_k(\mathbf{x}|\theta)$; label according to choice of $Z$ to form $D_+$.
Let $D = D \cup D_+$.
Train $\mathcal{M}$ on $D$.
**Return** $\mathcal{M}$

---

to limited training data. In this work, we evaluated four separate algorithms using a dataset with two orders of magnitude more samples, of which 194 had been previously identified as contaminated by clinical judgment of physicians. We reported 86.4% sensitivity and 99.9% specificity using a random forest classifier, however steps were not taken to account for the extreme class imbalance in the training set ($p = 1.39\%$). In particular, the results for the random forest sensitivity (86.4%) could not be reproduced reliably and in a stable manner. Additionally, we used the synthetic dataset only to demonstrate the scalability of our approach at the time.

## 2 METHODS

### 2.1 DATA

The total dataset consisted of 13,945 test results collected at the Robert J. Tomsich Pathology and Laboratory Medicine Institute of the Cleveland Clinic.[1] Of these samples, 194 were labeled as contaminated ($p = 1.39\%$) by a team of laboratory experts from the Insitute. The dataset was randomly split into training, validation, and test sets. All models were fit on an augmented version of the training set, with hyperparameter selection based on validation set performance and final performance measured on the test set.

### 2.2 MODELING

In this work, we use a data augmentation strategy that allows for stronger conclusions to be drawn and markedly improves model performance. Specifically, we train a Bayesian mixture model[2] with Gaussian components on contaminated samples, then sample from the mixture model to even out the class imbalance in the training set. Our exact procedure for data augmentation can be found in Algorithm 1.

We evaluate the approaches mentioned in section 1.1 as baselines under this data augmentation strategy, and then design an updated neural architecture. Included in this architecture is a soft attention mechanism applied to the input layer, allowing physicians some insight into how the model is attending to each analyte to determine contamination status. Hyperparameters for the neural network were tuned manually, while Bayesian optimization was used to find hyperparameters for each of the other learning algorithms.

## 3 RESULTS

Table 1 shows an evaluation of our model relative to previous work. All results are reported on the same held-out test set, which contained no synthetic data. All models were trained using the augmented training set described in Section 2.1 and Algorithm 1 to allow for a faithful comparison. In particular, the validation and test sets contained no synthetic data, and each model used the same validation and test set.

---

[1]This is the same dataset used in Mancuso et al. (2017), prepared in a manner compliant with HIPAA regulation.
[2]We also trained a standard mixture model, but its performance was much weaker due to the limited number of contaminated samples in our training set ($n = 149$).

Table 1: Best performing models (by F1 score).

| Algorithm | F1 Score | Precision | Recall | Specificity |
|---|---|---|---|---|
| Published Guidelines | 0.2372 | 0.4666 | 0.1590 | 0.9971 |
| CART | 0.6461 | **0.9545** | 0.4884 | 0.9841 |
| Random forest | 0.8076 | **0.9545** | 0.7 | 0.9935 |
| **Attention network** | **0.9333** | 0.9130 | **0.9545** | **0.9986** |

Table 2: Confidence intervals at 95% confidence. Computed over 80 random seeds for random forest, 30 seeds for attention network.

| Algorithm | F1 Score | Precision | Recall | Specificity |
|---|---|---|---|---|
| Random forest | (.776, .783) | **(.959, .965)** | (.651, .660) | (.991, .992) |
| **Attention network** | **(.868, .892)** | (.797, .838) | **(.954, .958)** | **(.996, .997)** |

In general, we find that our data augmentation strategy significantly improves results across all models. Our decision tree baseline performs significantly better than its predecessor in Baron et al. (2012), demonstrating that their 30-day mean glucose feature is not strictly necessary. The random forest baseline from Mancuso et al. (2017) also shows a marked improvement in stability as shown by the confidence intervals above. Our neural approach outperforms all previously proposed models when they are evaluated under similar conditions, with an absolute F1 score improvement from 80% to 93% over previous state-of-the-art approaches.

## 4 DISCUSSION

Our results demonstrate a clear need for improvement in how automated blood test contamination detection is currently performed in laboratories around the world. Detection with rules coming from published guidelines are incapable of performing adequately in this context. They are required to minimize false positives due to domain-specific requirements in laboratories, but are not sophisticated enough to do so without missing a large majority of contaminated samples. In contrast, learned algorithms offer a clear advantage when a few logistical concerns are dealt with – in particular, when the strong class imbalance is overcome and when a reasonable amount of data is available.

Additionally, the attention network allows for interpretability through visualization of the attention weights (see 4). The random forest afforded some interpretability through variable importance, but variable importance scores are not sample-specific. The attentional mechanism allows for case-by-case insight into the model's decisions, which is critical in a clinical setting.

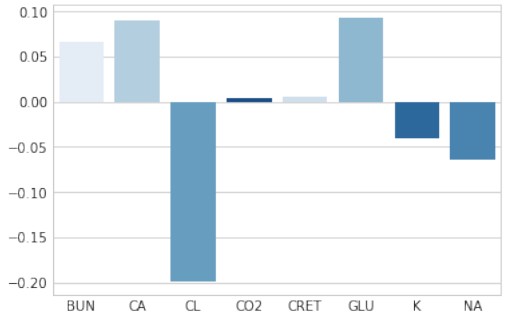

Figure 1: Attention weights for a single contaminated sample.

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
