# OpenReview forum: "Learning to Classify Contaminated Blood Tests: A Neural Approach"
_ICLR.cc/2018/Workshop — Reject_

### Official Review · AnonReviewer1 · 2018-03-07
**An important practical problem, but missing context**

**Rating:** 4
**Confidence:** 4

**Review:**

This paper presents a method for automatically classifying contaminated blood tests, which is an important practical problem for labs that run many tests.

This paper shows, basically, that in cases where data is very imbalanced, doing something about this helps. This is completely unsurprising, and there is no mention at all of class imbalance, or of the very large body of literature that exists on this topic. The point here, I realize, is not fancy new methods per se, but the authors also fail to make a compelling case regarding how this technology would be used.

To address relatively severe class imbalance, the paper includes a 'data augmentation' strategy, which entails fitting a mixture model over a subset of the data, although I think specification of this in Alg 1 is a bit vague. In particular, Z is drawn from "{X_{pos}, X_{neg}} suuch that |Z| = min(|X_{pos}|, |X_{neg}|)" -- this notation is unclear. Is Z a balanced sample? A simple IID sample of size Z? Regardless, the idea is to use this to generate pseudo-positive instances, thereby balancing the dataset. This seems very much like a probabalistic version of SMOTE, although the authors --- unfortunately --- do not mention this approach.

More should have been said about the features used here: A table describing these would be welcome. This is especially true given the focus on the application. Figure 1 is meaningless without description of the features.

In any case, it is not clear to me what the 'attention' layer is doing here. Attentive conditioned on what? What is the attention distribution induced over? If just the input features, isn't this just a linear layer?

Curiously, the authors do not compare against a simple logistic regression baseline trained in a way that accounts for imbalance (via down or upsampling, or SMOTE, or even class weighting, all standard tricks; the authors could also compare to the proposed strategy of course).

---

### Official Review · AnonReviewer2 · 2018-03-11
**Interesting application, more baseline results would improve the paper**

**Rating:** 6
**Confidence:** 4

**Review:**

This paper applied a neural attention network to identify contaminated blood tests samples. Since there exists great class-imbalance in the dataset, the training set was augmented by sampling from a fitted Bayesian mixture model. Experimental results showed the proposed approach outperformed 1) published guidelines; 2) CART; and 3) random forest w.r.t. F1 score. Overall, I think the application is interesting. However, I also have some questions on the details.

First, is it true that there were only 8 features in the dataset? Since the dataset is low-dimensional, more baseline approaches would provide better motivation for the proposed approach, such as vanilla feed forward neural network, boosting, support vector machine, etc.

Second, did the authors run some statistical tests, such as goodness-of-fit test, one-sample test, to confirm the trained Bayesian mixture model. which was used for augmenting the contaminated samples, has small overlap with the other class?

Third, instead of presenting the data augmentation details in Algorithm 1, it might be better to present the details of the proposed neural attention network instead.

---

### Official Review · AnonReviewer3 · 2018-03-11
**Important application of machine learning, yet limited relevance for ICLR**

**Rating:** 3
**Confidence:** 4

**Review:**

This report concerns detection of contamination in blood samples.  The contaminated class is undersampled. Authors suggest data augmentation to "even out" the undersampling. A "bayesian mixture model" is suggested.

Pros
* important problem
* representations for data augmentation is relevant to ICLR

Cons
*Severe lack of references for the models applied;
* therefore methods are incomprehensible
* the conclusion irrelevant to ICLR

---

### Decision · Program_Chairs · 2018-03-20
**ICLR 2018 Workshop Acceptance Decision**

**Decision:**

Reject

**Comment:**

Based on the reviews, this paper has not been accepted for presentation at the ICLR workshop. However, the conversation and updates can continue to appear here on OpenReview.